# Learning Optimal Test Statistics in the Presence of Nuisance Parameters

**Lukas Heinrich**

Technical University of Munich

E-mail: `lukas.heinrich@cern.ch`

**Abstract.** The design of optimal test statistics is a key task in frequentist statistics and for a number of scenarios optimal test statistics such as the profile-likelihood ratio are known. By turning this argument around we can find the profile likelihood ratio even in likelihood-free cases, where only samples from a simulator are available, by optimizing a test statistic within those scenarios. We propose a likelihood-free training algorithm that produces test statistics that are equivalent to the profile likelihood ratios in cases where the latter is known to be optimal.

## 1. Introduction

Statistical data analysis in the natural sciences is most often founded on a probabilistic modelling of the underlying data-generating process, where $p(x|\theta)$ denotes the probability of experimentally observing data $x$ given a set of theory parameters $\theta$. Inference aims at assessing the theory space in light of the observed data in order to estimate points or intervals in this space that are compatible with the data as well as test hypotheses for data-driven decision-making. In frequentist statistics the main tools for these tasks are estimates based on the well-developed methodology of maximum-likelihood estimation, confidence intervals construction and test statistics. In a Bayesian context, most inference tasks derive their results from methods that aim to compute posterior densities of the form $p(\theta|x)$. A major problem for both approaches, however, are *likelihood-free* settings, i.e. experimental situations where samples $x \sim p(x|\theta)$ are available but evaluating the likelihood $p(x|\theta)$ is computationally intractable. The field of *likelihood-free inference* thus aims to develop methods that allow us to still perform the desired inference tasks without requiring explicit evaluation of the model. High-Energy Physics data analysis is a prominent example of such a likelihood-free problem, which appears due to a rich, but unobservable evolution of the original particle collision through many latent intermediate states $z_i$ culminating into a high-dimensional measurement $x$. While the evolution probability itself is $p(x, z|\theta)$ is tractable, the model of the observable data $p(x|\theta) = \int dz \, p(x, z|\theta)$ is not. Classical approaches to *likelihood-free inference* often use simulation and summary statistics $f(x)$ to derive a low-dimensional approximate model $\hat{p}(f(x)|\theta)$ to which the standard methodology can then be applied. More recently a new breed of methods are developed that aim to use machine learning to eschew an explicit approximation of the statistical model, in favor of directly targeting only the model-derived quantities required for inference. In this work we add to this program by presenting a method to learn a test statistic with *best average power*. For models which lie in the asymptotic regime, this is equivalent to the *profile likelihood ratio* test statistic, which is a key quantity in frequentist data analysis for models that incorporate systematic uncertainties through nuisance parameters.

## 2. Related Work

Likelihood-free and simulation-based inference using machine-learning methods are growing field of research, where inference approaches for both Bayesian and frequentist settings are studied [1, 2]. The present work is most closely connected to the `carl` method of calibrated binary classifiers [3], where parametrized networks $t(x|\theta, \theta_0)$ are trained that are shown to be equivalent to the likelihood-ratio test statistic $p(x|\theta)/p(x|\theta_0)$. It is also closely connected to the `ACORE` [4] of likelihood-free hypothesis testing, but in our approach, we target particular proposal distribution designed to asymptotically recover the profile-likelihood through optimization. In addition, other approaches aim for parametrizing just a subset of model parameters to train an optimal observable but without using the resulting classifier as a direct proxy of a test statistic [5]. In our approach we target a test statistic that differentiates between parameters of interest $\mu$ and nuisance parameters $\nu$ and the resulting network is only parametrized by $\mu$ as a result. In a larger context this work is part of the program of exploring differentiable programming for high-energy physics inference [6, 7, 8] as it provides an approximation of a key inference-level quantity which is differentiable with respect to upstream analysis parameters.

## 3. Likelihood Ratio Tests for Nested Hypotheses

### 3.1. Frequentist Hypothesis Testing

Frequentist hypothesis testing and interval estimation aims at analyzing the data $x$ through the lens of a set hypotheses $H_i$. In binary testing the goal is to assess whether to reject a null hypothesis $H_0$ in favor of an alternative $H_1$. Tests are based on the comparison of the observed data to the sampling distribution $p(x|\theta_i)$ of the data under various theories $\theta_i$. Through an analysis of these distribution a *rejection region* $\omega$ is defined in the data space and $H_0$ is rejected if the observed data lies within that region. Practically, such regions are implicitly defined as the regions where a chosen *test statistic* $t(x)$ exceeds a threshold value $t_0$: $\omega = \{x|t(x) > t_0\}$ and the choice of $t(x)$ directly affects the performance characteristics such as the achievable *power* $\beta = p(x \in \omega|H_1)$ of the test at a given test size $\alpha = p(x \in \omega|H_0)$ . The search for *optimal* test statistics is therefore a priority in frequentist statistics.

### 3.2. Optimal Test Statistics

For simple hypotheses, where hypotheses correspond uniquely to a specific parameter points, the Neyman-Pearson Lemma [9] states that the universally most powerful (UMP) test is given by the (log-)likelihood ratio

$$t(x) = -2 \log \frac{p(x|\theta_0)}{p(x|\theta_1)}, \tag{1}$$

where $\theta_0$ and $\theta_1$ are the parameters of the null and alternative hypothesis, respectively. Composite hypotheses, where hypotheses correspond to sets in parameter space, generally do not admit UMP tests. However optimal tests can be found in more restricted cases. An important scenario is that of *nested hypotheses*, where the null hypothesis is completely contained in the alternative. A recurring setup is one where the full $n$-dimensional parameter space $\theta \in \mathbb{R}^n$ is partitioned into $k$ *parameters of interest* $\mu = \theta_{1:k}$ and $n - k$ *nuisance parameters* $\nu = \theta_{k+1:n}$. The null hypothesis is then identified as a subspace on which the parameters of interest assume a certain value $H_0 = \{(\mu, \nu)|\mu = \mu_0\}$. Wald [10] has shown that in this case the *profile likelihood ratio* test statistic given by

$$t_{\mu_0}(x) = -2 \log \frac{p(x|\mu_0, \hat{\hat{\nu}})}{p(x|\hat{\mu}, \hat{\nu})} \tag{2}$$

has a number of optimal properties in the asymptotic regime for any $\mu_0$. Here, $\hat{\mu}$ and $\hat{\nu}$ denote the global maximum-likelihood estimates, while $\hat{\hat{\nu}}$ denotes the maximum-likelihood estimate when

the search space is restricted to the null hypothesis set. In particular it can be shown to exhibit *best average power* under a suitable definition of the term, which we discuss in the Section 4.1.

*3.3. Likelihood-Free Learning of Test Statistics*

An immediate corollary of the optimality properties of test statistics $t(x)$ such as the likelihood ratio and the profile likelihood ratio is that they can be found by optimizing the parameters $\phi$ of a high-capacity function approximation $s_\phi(x)$, such as a neural network, with respect to a loss function which is minimized by the sought-after statistic $t(x)$. If the solution is unique, the approximation $s(x; \phi)$ will naturally converge to $t(x)$ with sufficient training. If a family of solutions exist, the optimization may converge on any member of such a family. In particular, evaluations of test statistics that rely on tail probabilities such as power and size, are invariant under a bijective variable transform and thus a solution $s(x)$ may be related to $t(x)$ through a bijective transform $t = f(s)$. If an external approximation $t_{\text{ext}}(x)$ of $t(x)$ is available, for example through e.g. explicit density approximation of $\hat{p}(x|\theta)$, the bijective calibration function $f$ can be found between the two through e.g. isotonic regression on $(s_i, t_{i,\text{ext.}})$ pairs. The procedure above yields a *likelihood-free* approximation of (a bijection of) $t(x)$ if the training process only requires samples from the probability models $x \sim p(x|\theta)$. For the simple hypotheses, the described approach is captured in the "likelihood ratio trick" of binary classifiers trained on the binary cross-entropy as a loss function [3]. The loss is minimized by a classifier that is monotonically related to the likelihood-ratio. We will now consider the likelihood-free learning for complex hypotheses and connect the results to asymptotic theory.

## 4. Learning Optimal Test Statistics for Nested Hypotheses

*4.1. Best Average Power Statistics*

We can extend the approach beyond simple hypotheses to find optimal test statistics $s(x; \mu_0)$ indexed by the values of the values $\mu_0$ assumed by parameters of interest for the null hypothesis set $H_0$ by training a parametrized neural network $s_\phi(x; \mu_0)$. As no UMP tests are available a metric through which to assess optimality needs to be defined. We choose to optimize for a measure of "best average power" as described by Wald [10], which we briefly recapitulate below. A reasonable approach is to define "best average power" with respect to "equally distant" alternatives. For example in the 1-dimensional case with a single parameter of interest $\mu$ and no nuisance parameters, a test of $H_0 = \mu_0$ is desirable that has similar power for alternatives $\mu_\pm = \mu_0 \pm c \cdot I_{\mu\mu}^{-1}$, that is for alternatives at a distance $c$ as measured within the Fisher metric. Generalizing to the case of nuisance parameters and higher dimensions we pick a given $\theta_0 \in H_0$ and first consider the a $k$-dimensional hyper-surface spanned by $n - k$ linear equations

$$\gamma_{ij}\theta^j = d_i(\theta_0), \text{ with } i = k+1 \ldots n, \ j = 1..n, \tag{3}$$

where the constants $\gamma_{ij}, d_i$ are chosen such that $\theta_0$ lies within that surface. This surface slices through the parameters space and have dimension $k$, which corresponds to the dimensionality of the parameters of interest. Within this surface of parameter points, we can now consider alternatives that are equidistant with respect to the $k$ parameters of interest to $\theta_0$, by computing the distance using the Fisher metric at $\theta_0$:

$$(\mu^i - \mu_0^i)(\mu^j - \mu_0^j)I_{ij}(\theta_0) = c, \text{ with } i, j = 1...k \tag{4}$$

The surfaces $S_c(\gamma, d, c)$ are $k - 1$ dimensional (hyper-)ellipsoids lying in the linear hyperplane defined by the coefficients $\gamma$. To compute the average power within the surface $S_c$ a density $p(\theta_{\text{alt}}|S_c)$ is needed to define the integral

$$\beta_{\text{avg}}(s_\phi, s_0, S_c) = \int_{S_c} d\theta_{\text{alt}} \, \beta(\theta_{\text{alt}}, s_0, s_\phi) \, p(\theta_{\text{alt}}|S_c), \tag{5}$$

where $\beta(\theta_{\text{alt}}, s_0)$ is the power of the test based on the test statistic $s_\phi$ and a rejection region defined by $\omega = \{x | s_\phi(x) > s_0\}$. The metric Wald uses is one that is uniform over the ellipsoids $S_c$. In general, computing the density on the surfaces $S_c$ of alternatives, or sampling from it, requires knowledge of the Fisher information. The somewhat complicated construction of $S_c$ significantly simplifies for the case of a single parameter of interest: the surfaces are just the two points that lie at a fixed distance from the null $(\mu - \mu_0)^2 = $ const. and on a linear subspace where $\gamma^T \theta = $ const. holds. Thus in this case, detailed knowledge of the Fisher Information is not necessary.

With Wald's definition of surfaces of alternatives above we can define a global measure of "best average power". Given some densities over the space of parameter points within a null hypothesis $p(\theta_0 \in H_0)$, of surfaces "equidistant alternatives" $p(S_c | \theta_0)$ and finally of alternatives on those surfaces $p(\theta_{\text{alt}} | S_c)$, as

$$\beta_{\text{global}}(s_\phi, s_0) = \int \mathrm{d}\theta_0 \, \mathrm{d}S_c \, \beta_{\text{avg}}(s_\phi, s_0, S_c) p(S_c | \theta_0) p(\theta_0), \tag{6}$$

and aim for finding a test statistic $s_\phi$ that has best average power for any threshold value $s_0$. If there is a solution $s_\phi$ that is optimal for any choice of $s_0$, $S_c$ and $\theta_0$ simultaneously, it will also be optimal under *any choice of densities* $p(\theta_0)$, $p(S_c | \theta_0)$.

### 4.2. Likelihood-Free Optimization

We can transform our optimization goal of "best average power" across a surface $S_c$ onto a likelihood-free optimization by realizing that optimizing for $\beta_{\text{avg}}$, that is the average power across a density of alternatives $p(\theta_{\text{alt}})$ is equivalent to optimizing the power of a hypothesis test with null distribution $p(x | \theta_0)$ and simple alternative designed as a mixture model on the surface $S_c(\theta_0)$ given by

$$p_{\text{mix}}(x | S_c) = \int \mathrm{d}\theta_{\text{alt}} \, p(x | \theta_{\text{alt}}) p(\theta_{\text{alt}} | S_c). \tag{7}$$

By the Neyman-Pearson Lemma and the Likelihood-Ratio Trick, the test statistic with the optimal power for a test of $H_0 : p(x | \theta_0)$ and $H_1 : p_{\text{mix}}(x | S_c)$ can be found by optimizing the binary cross-entropy:

$$\mathcal{L}_{\text{BXE}}(\phi | S_c, \theta_0) = -\mathbb{E}_{(y, x \sim p(x | \theta_0)), p_{\text{mix}}(x | S_c)}[(y \log s_\phi(x) + (1 - y) \log(1 - s_\phi(x))], \tag{8}$$

where samples from $\theta_0$ are labeled $y = 0$ and samples from the mixture of alternatives are labelled $y = 1$. To optimize the global power from Equation 6 in a likelihood-free way we can now average over $S_c$ and $\theta_0$ to define a global loss $\mathcal{L}(\phi)$

$$\mathcal{L}(\phi) = \mathbb{E}_{S_c, \theta_0}[\mathcal{L}_{\text{BXE}}(\phi | S_c, \theta_0)] \tag{9}$$

given choice of densities $p(\theta_{\text{alt}} | S_c), p(S_c | \theta_0), p(\theta_0)$. The final optimization procedure is simple: for each minibatch, sample $\theta_0$ and a random set of alternatives from $\theta_{\text{alt}}$ from a random surface $S_c(\theta_0)$, evaluate the neural network on data $x \sim p(x | \theta_i)$ and compute the average of the binary cross-entropy losses for each alternative, where samples from $\theta_0$ are labeled $y = 0$ and any samples from any $\theta_{\text{alt}}$ are labelled $y = 1$. With such a procedure and average loss definition, the parameters $\phi$ can be optimized via standard stochastic gradient descent. The algorithm is summarized in Algorithm 1.

### 4.3. Inference and Asymptotic Alignment with the Profile Likelihood Ratio

In general the test statistic found through the training procedure above depends on the choice of densities $p(\theta_{\text{alt}} | S_c), p(S_c | \theta_0), p(\theta_0)$. Furthermore, the sampling distributions $p(s | \theta)$ under a member of the null hypothesis set $\theta \in H_0$ or its encompassing alternative set $\theta \in H_1$ is unknown.

---
**Algorithm 1** Training a Test Statistic with Best Average Power
---
**Require:** $\eta$: learning rate
**Require:** $\phi_0$: initial parameters
**Require:** $\theta \sim p(\theta)$, $\theta \sim p(\theta, S_c|\theta_0)$: sampling routines
 1: **while** not converged **do**
 2:     $\theta_0 = (\mu_0, \nu_0) \sim p(\theta)$                                                   ▷ sample null
 3:     $\theta_i = (\mu_i, \nu_i) \sim p(\theta, S_c|\theta_0)$                              ▷ sample alternatives
 4:     $(x_i, y_i) \sim p(x|\theta_0), p(x|\theta_i)$                    ▷ null: $y_i = 0$, all alternatives have $y_i = 1$
 5:     $p_i \leftarrow s_\phi(x_i; \mu_0)$
 6:     $L = \sum_{\text{null,alts}} L_{\text{BXE}}(y_i, p_i)$
 7:     $\phi_{i+1} \leftarrow \phi_i - \eta \nabla_\phi L$
 8: **end while**
 9: **return** $\phi_N$
---

This is no impediment for inference as the sampling distributions for any $\theta$ can be found easily by evaluating the learned test statistic $s_\phi$. Based on such empirical distributions a consistent frequentist inference procedure for hypothesis testing as well as point and interval estimation can be constructed.

Our choice of "best average power" as an optimization target, however, is not an accident and an interesting connection appears in the limit of asymptotic behavior. Due to the results to Wald mentioned in Section 3.2, we know that the *profile likelihood ratio* is the test statistic that has best average power for *any* surface $S_c$, $\theta_0$ and threshold value $s_0$ and thus will also minimize the global measure power $\beta_{\text{global}}$ as well as the loss defined in Equation 9. In case of models $p(x|\theta)$ which satisfy sufficient asymptotic properties, the likelihood-free search for a best average power test statistic $s_\phi(x; \mu_0)$ is expected to yield a test statistic that is one-to-one to the profile likelihood ratio. We explore this connection empirically below.

## 5. Experiments

We demonstrate the approach of implicitly learning a best average power test statistic two examples that have tractable $p(x|\theta)$ and are known to be within the asymptotic regime. This allows us to verify that the training procedure finds the optimal solution as we can independently compute the profile likelihood ratio. Furthermore, the calibration function relating $s_\phi(x; \mu_0)$ and $t_{\mu_0}(x)$ can be found through isotonic regression. On the one hand we study a synthetic Gaussian Example with fixed covariance matrix and on the other hand we study the 'on-off' problem measuring two Poisson processes with shared parameters. In each case we train neural network with three fully connected hidden layers and with width 100 and subsequent tanh activation. The final output is passed through a sigmoid activation and then evaluated through the binary cross-entropy as described above. The networks are trained with the Adam optimizer [11] with learning rate $\eta = 5 \cdot 10^{-5}$ and a mini-batch size of $n = 1000$. We implement the training in PyTorch [12] and provide the code to reproduce the figures on GitHub [1]. The examples are partially chosen because closed-form solutions of the profile likelihood ratio are available, which facilitates the comparison with ground-truth and the derivation of calibration functions. We follow a simplified sampling scheme for the null and alternative hypotheses suited for one dimension, where we choose $\gamma_{11} = 0$ and $\gamma_{12} = 1$, such that the null and the alternatives to the left and right of the null have the same nuisance parameter value $\nu$, which is sampled according to a uniform distribution. The distance $(\mu - \mu_0)^2 = $ const. is also sampled uniformly for each minibatch.

---

[1] See the repository at `https://github.com/lukasheinrich/learning_best_average_power`

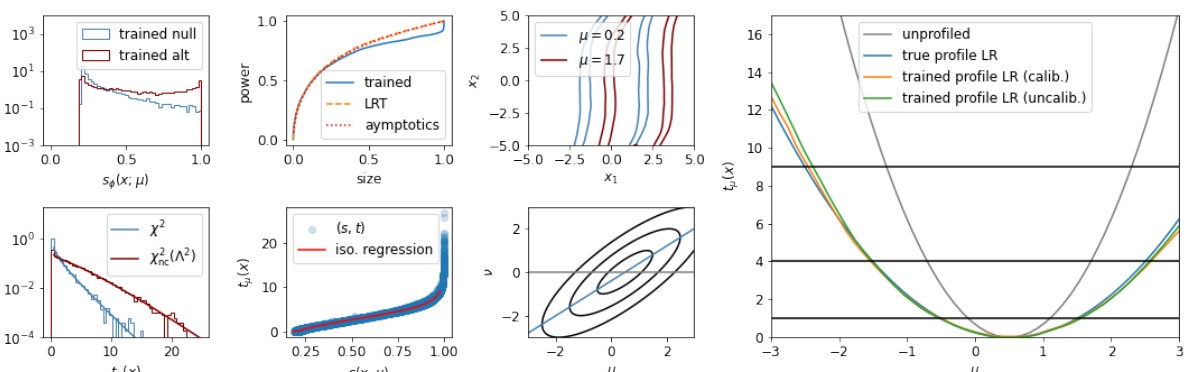

Figure 1: Results of the described method for a Gaussian example. On the left-most column the trained test statistic distribution $s_\phi(x; \mu_0)$ (top) and the true profile likelihood ratio (botttom) that closely follows asymptotic theory. To the right, the ROC curves (top) of the trained classifier, likelihood ratio, and asymptotic theory are shown to match well. The joint distribution (bottom) suggests a monotonic relationship between the two. In the middle, the learned test statistic is shown in data space for two parameters values (top) and the true likelihood is shown for a example observation together with the target profile. The right pane compares ground truth and trained profile likelihood values as a function of $\mu$.

### 5.1. Gaussian Example

In this example we consider a bivariate Gaussian Model with an arbitrary mean $\theta = (\mu, \nu)^T$ but fixed covariance $\Sigma$

$$p(x_1, x_2 | \mu, \nu) = \mathcal{N}(\begin{bmatrix} x_1 \\ x_2 \end{bmatrix} | \begin{bmatrix} \mu \\ \nu \end{bmatrix}, \Sigma)$$

As asymptotic theory describes models in which maximum likelihood estimates achieve unbiased normal distributions and their variance is described by the Cramér-Rao bound, this example is a good proxy for such models and we expect the optimal solution found in training to reproduce the profile likelihood ratio (modulo bijection) to a very high degree of precision. Moreover, there is a trivial relationship between data and the model parameters. The results are shown in Figure 1. By comparing the ROC curves of the two test statistics, we see that they are very compatible, which suggests a bijective relationship between the learned test statistic and the true profile likelihood ratio. The bijection is evident in the joint distribution of the (known) profile likelihood ratio and the learned statistic from which we can extract a calibration function through isotonic regression as implemented in `scikit-learn` [13]. We can recast the learned statistic $s_\phi(x; \mu_0)$ into $t_{\mu_0}$-like units through either the calibration function or through percentile-matching to the $\chi^2$ distribution to compare the trained classifier to the true profile likelihood ratio. As shown in Figure 1, both the network is an excellent approximation of the true profile likelihood also in the uncalibrated case, where no ground-truth information is used.

### 5.2. On-Off Problem

In this example we study the classic "on-off" problem [14] of simultaneous measurements of two Poisson processes:

$$p(x_1, x_2 | \mu, \nu) = \text{Pois}(x_1 | \mu s + \nu b)\text{Pois}(x_2 | \nu \tau b), \tag{10}$$

where the hyperparameters $s, b$ indicate the nominally expected signal and background counts. The hyperparameter $\tau$ describes the relationship in measurement time between the two processes. The signal strength $\mu$ is the parameter of interest while the scaling factor for the background $\nu$ is

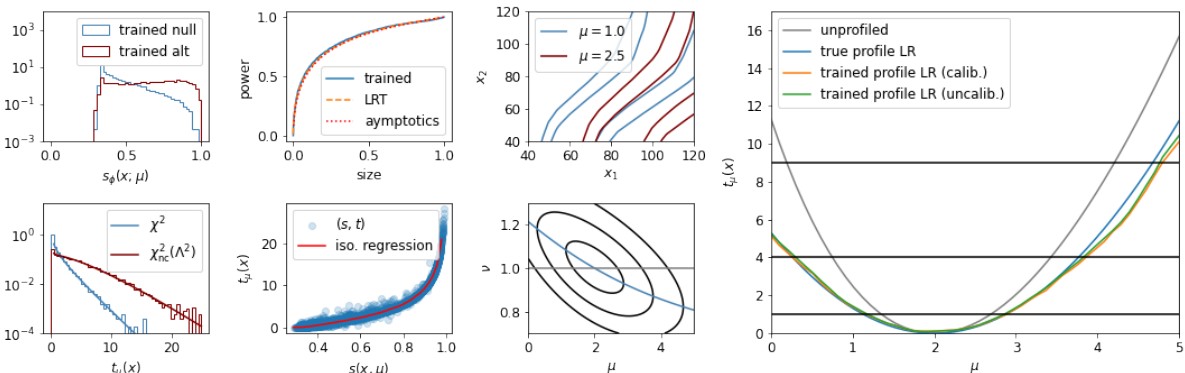

Figure 2: Results of the described method for the on-off problem.

a nuisance parameter. This model deviates from a pure Gaussian setup and introduces a more complicated relationship between data and parameters. The results are shown in Figure 2. With hyperparameters set at $s = 15$, $b = 70$, $\tau = 1$ the model is comfortably in the asymptotic regime but with $\tau \sim 1$ significant deviation the profiled values of the nuisance parameters is expected as one moves away of the maximum-likelihood estimate. As in the Gaussian case, the neural network directly approximates the profile likelihood ratio to very high degree of precision.

## 6. Conclusion

We have demonstrated that we can find powerful test statistics in a likelihood-free way through optimizing for average cross-entropy, which was shown to be equivalent to converege to best average power statistics. In the limit, where the intractable model $p(x|\theta)$ displays asympptic behavior, the learned test statistic was shown to be equivalent to the profile-likelihood ratio. This equivalence was empirically shown to hold to a very high degree of precision in two example cases that are known to behave asymptotically. The trained classifier can be directly used for frequentist inference tasks such as point and interval estimation. During inference, no actual data-dependent minimization needs to be performed and as such the network $s_\phi(x; \phi)$ is an *amortized inference* tool, in which a large one-time cost, i.e. the training, is traded off against fast instance-level inference. It's interesting to note that in cases where asymptotic assumptions do not hold this procedure may produce test statistics that perform better with respect to the average power metric than the profile likelihood ratio as the latter may not be the optimal solution anymore.

## Acknowledgements

LH thanks Allen Caldwell, Kyle Cranmer, Nathan Simpson, Alexander Held and Michael Kagan for fruitful discussions and comments on the manuscript. LH is supported by the Excellence Cluster ORIGINS, which is funded by the Deutsche Forschungsgemeinschaft (DFG, German Research Foundation) under Germany's Excellence Strategy - EXC-2094-390783311. Our code makes use of PyTorch [12], Numpy [15], SciPy [16], Scikit-Learn [13], Matplotlib [17] and Jupyter [18].

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
