# Peer review of "Learning Optimal Test Statistics in the Presence of Nuisance Parameters"

_SciPost Physics_

## Round 1 · Referee Report · Anonymous (Referee 1) · 2024-3-27

Strengths

  1. The paper documents a novel method for learning optimal test statistics in the face of non-simply hypothesis testing, by extending the likelihood-ratio trick using a notion of average power.
  2. The paper makes use of example (toy) studies in which the exact likelihood can be determined to showcase the performance of the method.
  3. The code to perform these studies has been provided and is available for the reader.

Weaknesses

  1. The explanations in some cases are rather hard to follow and the reader is left often to fill in gaps based on some guess work. Essentially the paper assumes a level of expertise/familiarity with concepts that the audience might not possess.
  2. the studies that are performed are somewhat incomplete making some of the conclusions that are drawn hard to justify completely.

Report

Thanks for this interesting paper draft on the use of ML to learn optimal test statistics in nested hypothesis, and the presence of nuisance parameters. Overall I find the concept and method discussed in the paper very interesting and it clearly shows promise. However, I think that the paper in its current form is far from being publishable and even a moderately expert reader would struggle to understand everything that is presented.

I have a number of comments below, ranging from minor edits / questions to rather substantial suggestions for additions, that I think will improve the paper. Please do consider them for resubmission as I believe this work deserves to be published.

Regards,

Your referee

General Comments:

  • I find in several places that the notation is hard to follow. This is particularly true in section 4.1 when discussing the tests of average power. I would recommend referring back (more often than currently) to the simple 1D or even a 2D case where the form of things like Sc and examples of p(.|Sc) can be given explicitly. This would really help the reader to understand conceptually how the test works in concrete cases. In Section 5, then each of the cases demonstrated could be explicit about what is used for those terms.

  • The figures that are intended to demonstrate that the procedure works as intended and their explanation is very poor in the paper. In Figure 1 for example, the caption refers to a “middle column” for a 4 column figure so the reader isn’t even exactly sure where to look. I have more comments specifically about this below but generally I don’t find the explanations about the demonstrations very convincing from what is written in the paper.

  • In a couple of places, there are terms that the intended audience (presumably HEP data analysts given that this example is used as a case where the integration is intractable) are not likely to know. The one I noted is “isotonic regression”, which seems to be an important tool used to calibrate the approximation to yield the desired test statistic. It would be useful to explain this in a few lines just so that the reader knows what is meant by this term. The same goes for ROC curves (though here its true a HEP person would know, but perhaps it could still be spelled out for the general audience).

  • Generally there are quite a few errors in the grammar (missing or extra words in sentences), only some of which I point out below.

Requested changes

  1. Section 1, pp1, L3 - I would change “aims at” to “is the process of” since inference is the “best” procedure we have for exploring this space accounting for the data. There’s a couple of other “aims at”/“aims to” in the paper which can be similarly modified.

  2. Section 1, pp1 - When you mention the Bayesian context, you give an expression for the posterior but in the sentence after, you state that both Bayesian and Frequentist approaches make use of the likelihood function. This is true, however it would probably be clearer to the reader if you explain that the posterior is proportional to the same likelihood that appears in the frequentist case, just to hammer this point home.

  3. Section 2, pp2, L1 - methods are PART OF A growing field …

  4. Section 3, pp3 - I would add “the null hypothesis set DEFINED BY M0” just to make it clear what the null hypothesis is.

  5. Section 3.3, pp3 - Are s_{\phi}(x) and s(x;\phi) the same or is one explicitly meant to be an approximation of the other? If the latter, which one is supposed to be an exact bijection with t and which is the approximation for which a bijection can be found? I don’t think i’ve understood the subtly here so a but more explanation would really help in this paragraph.

  6. Section 3.3, pp3 - about halfway down this section you say “for example though e.g”, which is for example twice. One can be removed

  7. Section 3.3, pp3 - here is the first time that isotonic regression appears so I think it could be explained or at the very least a reference found for it. Otherwise, the reader is assumed to know exactly what is being done here, which i don’t think should be the case.

  8. Section 4.1, pp4 - after Eon 6, you state that the optimality holds for any choice of densities. Is the same true for another choice of metric (for “equidistant”)? Would it matter if I chose something other than the Fisher metric or would that just be essentially redefining what we mean by “average optimality”?

  9. Section 4.2, pp4 - after En 8 I think would be a good place to insert a specific example of a 1D or 2D case and what Sc, P(x|Sc) would look like in that case. Even though the BXE is pretty well understood, I think it will help contextualise for someone that is used to dealing with it in the simple case of two simple classes (H0 vs H1) where H0 is really a simple hypothesis.

  10. Section 4.3, pp5 - I had some trouble understanding the notation of algorithm 1. In steps 5 and 7, does this just mean that for the specific sample xi, one evaluates the summary s\phi for xi to yield the term pi that enters the BXE (right?). Sorry if this is obvious, just wanted to make sure i have understood this.

  11. Section 4.3, pp5 - In the text just after algorithm 1, could you give an example of how the point and interval estimation can be constructed. Do you just mean for example that once s\phi is learned, one can either calibrate to find t (and thereby use the standard point estimate from where t=0 and 68% interval in the range t<1 for example), or through a neyman construction do something similar? If you could elaborate on this, or provide an example (maybe forward referencing to section 5) it would really help understand what is meant here.

  12. Section 5, pp6, Figure 1 - As mentioned in the General comments, I think its very hard to understand some of these figures. You should perhaps label each panel (a), (b), (c) etc, to allow you to refer to each sub figure in the caption. The 3rd column is not really explained - what are the red and blue lines exactly? what are the contours and diagonal line in the bottom part ? For the right most figure, I don’t think “calib” or “uncalib” are defined anywhere in the text, nor is “unprofiled”, nor is “profile LR” (is it the same as LRT?) etc. You should use the same labelling as elsewhere otherwise a reader is just guessing. All of the above goes the same for Figure 2, but more so since the caption tells us nothing really in that case.

  13. Section 5.1, pp6, For this first model, by fixed covariance, you mean it is known right? You should specify that. I had a couple of questions to clarify what is being done here: a) When comparing the ROC curves, there is some visible difference between the trained and true curves. Is this something that can be overcome with more training data/ more complex network etc or is it inherent in the method somehow. b) You are learning the profiled likelihood ratio (here provide over \nu) which of course implicitly depends on x1 and x2. You could mention exactly what the values x1 and x2 are for your right figure in Figure 1, even if one can figure it out by starting at the lower plot in the 3rd column. Is the (rather small) spread in the distribution of s(x,\mu) vs t(x) coming from different values of x1, x2 and can one assume that for any values, this red line would be the same? i.e is the calibration a fixed thing or does it depend in the end on the dataset observed. c) one could imagine training the full likelihood ratio (ie before profiling) which of course loses the advantage that it is an amortized inference tool since then the user has to go and do some profiling on top, but presumably the advantage is that this calibration curve is less dependent on the data (depending on the answer to b).

  14. Section 6, pp7 - You say that you have demonstrated being able to find powerful test statistics (which I agree with I think), but I wonder if you considered also comparing the power one obtains with the case when the alternate is exact. Somehow, the tradeoff is that by considering a set of alternatives and maximising average power, you would lose some power compared to a specific test with exact H0 and H1, but its not clear how much power in the end you would lose. I would be interested to see such a study. This could be done for a range of alternatives to really show how one gains from this approach where the average power is maximised over that set of alternatives (ie a test optimised for H1 would do worse at H1’ and vice versa in the simple-vs-simple).

  15. Section 6, pp7 - the last sentence is very intriguing but as far as i could tell, is not backed up by anything in the paper. Is there a reference for this or is this more of a speculative statement? If its beyond the scope of this paper, i’d be tempted to remove it or add some discussion earlier (say in Section 4) to back it up. It might already be in there somewhere but its not obvious to me.

---

## Editorial Decision

awaiting_resubmission